# Effectiveness of Different Modalities of Remote Online Training in Young Healthy Males

**DOI:** 10.3390/sports10110170

**Published:** 2022-11-02

**Authors:** Michael Daveri, Andrea Fusco, Cristina Cortis, Gabriele Mascherini

**Affiliations:** 1Department of Experimental and Clinical Medicine, University of Florence, 50134 Florence, Italy; 2Department of Human Sciences, Society and Health, University of Cassino and Lazio Meridionale, 03043 Cassino, Italy

**Keywords:** livestreaming exercise, on-demand exercise, home exercise, supervised exercise

## Abstract

Since 2020 there has been an increase in demand for home workouts. Therefore, different ways of delivering distance training have been proposed to promote “stay active at home.” This study aimed to compare the effectiveness of three different training programs consisting of a total of 15 workouts (three sessions per week): supervised livestreaming (LS), unsupervised following a video recording (VR), and unsupervised following a written program (WP). Changes in anthropometric and cardiovascular variables, muscle fitness, and physical activity levels were evaluated. To provide a meaningful analysis for significant comparisons between small groups, mean differences (∆), 95% confidence interval (95% C.I.), and Cohen’s effect sizes (E.S.) were also calculated. The three training modalities increased physical activity levels, with an adherence rate of LS = 93.3%, VR = 86%, and WP = 74%. Although there was no reduction in body weight, waist circumference decreased by 1.3 cm (95% C.I. = −2.1, −0.5; E.S. = 0.170; *p* < 0.004). Furthermore, where LS, VR, and WP resulted in improvements in muscle fitness, only LS showed changes in cardiovascular variables, such as resting heart rate (∆ = −7.3 bpm; 95% C.I. = −11.9, −2.7; E.S. = 1.296; *p* < 0.001) and Ruffier’s index (∆ = −2.1bpm; 95% C.I. = −3.5, −0.8; E.S. 1.099; *p* < 0.001). Remote online training proved its effectiveness over a short period of time. However, supervised training proved to be the most effective, highlighting the importance of an experienced trainer.

## 1. Introduction

On 30 January 2020, the World Health Organization (WHO) declared the coronavirus pandemic a global emergency. In Italy on 9 March 2020, non-essential and strategic activities such as gyms, swimming pools, wellness centers, and clubs were suspended [1]. The coronavirus outbreak brought enormous challenges for the fitness training sector, with an increased use of technology and the development of new training strategies [2]. As the pandemic progresses, most activities have been moved online, including medical screening and participation in physical activity [2].

Performing sedentary behaviors for at least seven hours a day carries an independent risk factor for numerous chronic diseases and overall mortality [3]. Therefore, the promotion of physical activity is an action declared by the WHO, aiming for an active lifestyle that favors the benefits of physical training, thus reducing the consequences that physical inactivity entails [4].

During the pandemic, there was an increase in online searches for “home workout” [5]. Therefore, the use of the internet has been exacerbated as well as the use of technology, as individuals tried to stay active through applications, websites, videos, and social media, engaging in remote online workouts [6].

Previous studies showed some advantages regarding online training, showing how it can bring partial benefits compared to face-to-face training [6]. Other recent research shows the feasibility of online remote training for diseased populations, such as cancer and multiple sclerosis patients [7,8].

It seems that the presence of the instructor in the face-to-face mode brings advantages in terms of motivation and attention, which then translate into better fitness parameters [6]. However, the instructor’s supporting role during online training programs has not yet been studied. The hypothesis of this study is that the supervision of the instructor can provide greater improvements, even in remote online training.

It is conceivable that the use of technology in remote training could continue even after the end of the coronavirus pandemic; therefore, evaluating the effectiveness of different approaches can be helpful to choose the most effective solution to engage people in physical activity [9].

Considering the advantage of online training during the pandemic where social distancing is required, this study aims to investigate the effects of three different ways of remote online training on anthropometric and cardiovascular variables, muscle fitness, and physical activity levels, by evaluating their feasibility and effectiveness.

## 2. Materials and Methods

### 2.1. Participants and Study Design

The recruitment process involved the dissemination of the initiative among university students. Those who: (1) gave their availability, (2) guaranteed the achievement of the sample size, (3) met the inclusion criteria, and (4) signed informed consent were enrolled in the study. Twenty-one healthy males (23.1 ± 1.5 years; height 180 ± 0.1 cm; weight 73.4 ± 8.7 kg) were enrolled in this study. Inclusion criteria were: aged 18–25 years, no contraindications to exercise, access to the internet, and availability for approximately 2 h per day to spend in the training sessions.

Exclusion criteria were: aged under 18 or over 25 years, no access to the web platform used, recent injuries, practicing sports at competitive levels, unavailable during the 2-h training sessions.

The study was undertaken in accordance with the ethical standards laid down in the 1975 Declaration of Helsinki. The Institutional Review Board of the Department of Human Sciences, Society, and Health of the University of Cassino and Lazio Meridionale approved this study (approval No.: 9947; date: 14 April 2021).

Eligible subjects were divided into 3 groups (n = 7 for each group) following the model for randomized studies with parallel groups:

1. Livestream group (LS): the subjects received the instructions directly from the experienced trainer via livestream on a web platform. The experienced trainer explained each exercise; his direct intervention showed the correct execution, corrected any errors, motivated the group, and supervised the execution in terms of sets, repetitions, rest, and range of motion.

2. Video recording group (VR): the subjects received the instructions from a video recording uploaded on a web platform. The videos showed the different training sessions in terms of exercises, the correct execution, sets, repetitions, and rests. The experienced trainer did not participate live during the training sessions, and each subject had to train individually at any time of the day. However, the experienced trainer was fully available (phone call, instant message, etc) to respond to any requests from participants.

3. Written program group (WP): the subjects received instructions from a written training program. The program was first explained in detail, then sent and made available on smartphones for greater daily availability. The program showed the exercises through photos and descriptions of the execution and series, repetitions, and rest. Similar to the VR group, the experienced trainer did not participate directly during the training sessions, although they were available upon request.

The evening before the training day, a reminder was sent to all participants when the training session was uploaded to the group platform (link for LS, video recording for VR, and program sent by phone message for WP). In addition, to verify adherence to the training program, members of the VR and WP groups had to send a statement at the end of each completed workout, while for the LS group, the experienced trainer recorded the participation.

### 2.2. Procedure

The effectiveness of the 3 training programs was evaluated by comparing the pre-post training results of anthropometric evaluations, flexibility and balance, muscle strength, cardiovascular parameters, and weekly physical activity level. The experienced trainer performed assessments at each subject’s home to ensure that the test was performed correctly. Therefore, 3 attempts were granted for each test and the best result was considered for further analysis [10].

#### 2.2.1. Anthropometric Variables

Weight was measured to the nearest 0.1 kg and height to the nearest 0.1 cm (Seca 786 GmbH and Co. Hamburg, Germany). Body mass index (BMI) was calculated as the body mass divided by height squared (kg/m^2^). Waist circumference measures were made with a tape metric (Holtain Limited, Crymych, UK, 1.5 m Flexible Tape) at the narrowest level, or if this was not apparent, at the midpoint between the lowest rib and the top of the hip bone (iliac crest) [11].

#### 2.2.2. Muscle Fitness

Muscle fitness was assessed by testing for: (1) flexibility, (2) joint mobility, (3) balance, and (4) muscle strength [12].

1. The sit and reach (SR) test was used as a measure of the posterior extensibility of the body [13]. Measurements were taken as the distance from fingertips to toes during trunk flexion.

2. Shoulder flexion and extension tests measure the angles of flexion and extension of the shoulder and evaluate the possible retraction of the shoulder girdle muscles. The shoulder internal rotation test measures the angle of internal arm rotation from 90° abduction and 90° forearm flexion and allows evaluation of the external shoulder rotation functionality [14].

3. The one leg test measures the ability to balance in a single stance with eyes closed. This enables the evaluation of body sensory management [15].

4. The hand grip test estimates the overall static strength of the upper limb. The subjects were in a sitting position with the shoulders adducted and rotated in a neutral position, the elbow flexed at 90°, and the forearm in a neutral position. Three tests were carried out on each side [16]. The maximal push-up test was administered to evaluate upper-body muscle endurance by repetition until muscle failure or incorrect execution. For a correct push up, the subject had to keep his back straight, and at each stretch up the elbows had to lock [17]. Finally, the maximal plank test was administered to evaluate core muscle strength. Subjects had to stand prone with shoulders and elbows flexed at 90°, maintaining a straight line from head to toe without lowering the hips, keeping the neck in a neutral position [18]. Subjects were required to maintain the position for as long as possible and the time was recorded.

#### 2.2.3. Cardiovascular Variables

When the participants were seated and relaxed for at least 3 min, the resting heart rate was measured by pulse oximeter (CocoBear, Pulse Oximeter PR-10). Resting heart rate is an independent cardiovascular risk factor, and it could help to identify potential health problems [19]. For effort tolerance, the Ruffier test was used. This allows the evaluation of the subject’s recovery capacity from an effort. Each subject was asked to perform 30 squats in 45 s, followed by heart rate recording; after 1 min of rest, heart rate was measured again and the Ruffier index was estimated [20].

#### 2.2.4. Physical Activity Level

The international physical activity questionnaire (IPAQ) provided standard instruments that could be used to gain internationally comparable data on health-related physical activity [21]. The questionnaire, which queries time spent physically active in the last 7 days, comprises of 4 sections:

1. Vigorous physical activities: minutes per week. This refers to activities of hard physical effort which make breathing much harder than usual.

2. Moderate activities, but not walking: minutes per week. These activities refer to actions of moderate physical effort which make breathing somewhat harder than usual.

3. Walking: minutes per week and number of daily steps. This includes at work and home, walking to travel from place to place, and any other walking undertaken solely for recreation, sport, exercise, or leisure.

4. Time spent sitting, including reading, watching television, studying, and playing video games: minutes per day.

### 2.3. Training Program

The training program lasted 5 weeks with a frequency of 3 sessions per week (Monday, Wednesday, Friday) for a total of 15 sessions.

Each training session included 3 parts: warm-up with flexibility and range of motion exercises (20 min), whole-body muscle strengthening (45 min), and cool-down with static muscle stretching (15 min). The whole-body muscle strengthening part was based on exercises individualized to the participants’ ability, ensuring that participants could perform those exercises at home independently. Therefore, this mainly included bodyweight exercises, such as push-ups, squats, lunges, scapular adductions and planks, as these do not require special equipment. The program was based on the methodology training fundamental principles; the routine had a progressive load increase each weeks, had training load variety through different kinds of exercises, and training load specificity to improve participants’ strength levels and improve posture [22]. In order to verify the effectiveness of the 3 training programs, all 3 groups followed the same training program and the same progression of the load. During the first week, each group performed 3 sets of 12 repetitions with 60 s of recovery between sets for each exercise. During the second and third weeks, the sets were increased to 5, and in the fourth and fifth weeks the repetitions were increased to 15. The main muscles involved in the training program were the pectoral, back, thigh, shoulder, and core muscles, such as the pectoralis, latissimus dorsi, quadriceps, hamstrings, gluteus, deltoids, and abdominal.

### 2.4. Statistical Analysis

Stata statistical software version 14.2 (Stata-Corp, College Station, TX, USA) was used for statistical analysis. All variables were calculated by mean, standard deviation (SD), mean difference between pre-post (∆), and 95% confidence intervals (95% C.I.). The Cohen’s d effect size (E.S.) was calculated to determine the magnitude of effect. The E.S. was assessed using the following criteria: small < 0.20, medium < 0.50, and large < 0.80. Multilevel regression models (or hierarchical linear models) were performed to examine the effects of different training modalities on the subject’s strength, mobility, and cardiorespiratory fitness performances. Subjects were considered the random effect, whereas the training modalities and testing time (pre vs. post) were treated as fixed effects. The models were fitted using the maximum residual likelihood for accounting for the small sample. The repeated measures ANOVA method was used for computing the degrees of freedom of a t distribution, as subjects were tested before and after the training protocols. Subsequently, the contrast method was used to test whether the dependent variable (i.e., the Ruffier test and push-up test) meant each training modality and time were identical. The contrast method tests included ANOVA-style tests of the main effects used to make comparisons against the reference categories (i.e., pre vs. post; LS vs. VR vs. WP). The statistical significance of the main effect on the whole sample analysis was set at *p* < 0.05. Subsequently, Bonferroni post-hoc tests were used for multiple-comparison adjustments across all terms where significant main effects and interactions were found. After Bonferroni correction, statistical significance was set at *p* < 0.003.

## 3. Results

The remote online exercise program adherence rate in the three groups was LS = 93.3%, VR = 86%, and WP = 74%.

### 3.1. Changes after Remote Training on the Whole Sample

The effects of remote training on the whole sample are shown in Table 1.

#### 3.1.1. Anthropometric Variables

Based on the anthropometric parameters, the sample included normal weight subjects within the normal range with waist circumference. On average, there was no significant weight loss (pre 73.4 ± 8.7 kg, post 73.2 ± 8.6 kg), and therefore the BMI did not decrease either (pre 22.6 ± 2.3 kg/m^2^; post 22.5 ± 2.2 kg/m^2^). On the other hand, the waist circumference shows a significant reduction (pre 83.6 ± 6.0 cm; post 82.3 ± 5.7 cm; *p* < 0.004).

#### 3.1.2. Muscle Fitness

Muscle fitness parameters show significant increases in the muscle strength component evaluated in the push-up (pre 26.3 ± 10.1 rep.; post 34.0 ± 7.9 rep.; *p* < 0.001), plank (pre 90.8 ± 26.7 s; post 117.4 ± 30.4 s; *p* < 0.001), and handgrip (pre 88.7 ± 21.3 kg; post 95.6 ± 19.3 kg; *p* = 0.036) tests. For the joint mobility component of muscle fitness, only shoulder extension did not show significant differences (pre 60.5° ± 9.7; post 66.8° ± 8.8). However, the other joint mobility assessments show significant increases in shoulder flexion (pre 174.2° ± 7.0; post 185.5° ± 5.0; *p* < 0.001) and internal rotation (pre 56.4° ± 16.9; post 69.3° ± 10.1; *p* = 0.003). Flexibility and balance show improvements in the sit and reach test (pre 26.1 ± 8.6 cm; post 29.0 ± 7.2 cm; *p* = 0.018) and one leg test (pre 23.9 ± 18.0 s; post 55.5 ± 47.4 s; *p* = 0.03).

#### 3.1.3. Cardiovascular Variables and Physical Activity Level

Cardiovascular variables showed a significant reduction in resting heart rate (baseline 69.0 ± 7.2 bpm; post 66.3 ± 4.1 bpm) and Ruffier’s index (baseline 9.0 ± 2.2 bpm; post 7.9 ± 1.6 bpm). Physical activity levels also increased significantly (baseline 1639.6 ± 1531.6 METs; post 2991.4 ± 1715.3 METs).

### 3.2. Changes after Remote Training According to the Division into Groups LS, VR, and WP

For the variables that showed significant differences in the whole group analysis, an analysis was performed to verify the effectiveness of remote training for each delivery modality used (Table 2, Table 3 and Table 4).

No differences in delivery modalities of remote training were observed in:-waist circumference (LS: baseline 85.5 ± 3.6 cm, post 84.2 ± 3.5 cm, *p* = 0.012; VR: baseline 85.6 ± 7.1 cm, post 84.6 ± 6.9 cm, *p* = 0.080; WP: baseline 79.6 ± 5.5 cm, post 79.1 ± 5.3 cm, *p* = 1)-sit and reach (LS: baseline 25.7 ± 9.5 cm, post 29.1 ± 6.8 cm, *p* = 0.134; VR: baseline 23.3 ± 10.3 cm, post 26.9 ± 9.4 cm, *p* = 0.097; WP: baseline 29.4 ± 5.3 cm, post 31.1 ± 5.1 cm, *p* = 1)-shoulder internal rotation (LS: baseline 56.4° ± 17.9, post 71.4° ± 10.1, *p* = 0.011; VR: baseline 60.0° ± 19.1, post 69.6° ± 10.4, *p* = 0.445; WP: baseline 52.9° ± 15.6, post 66.8° ± 10.8, *p* = 0.025)-one leg test (LS: baseline 16.2 ± 9.5 s, post 49.0 ± 39.4 s, *p* = 0.274; VR: baseline 24.3 ± 19.0 s, post 43.9 ± 29.1 s, *p* = 1; WP: baseline 31.2 ± 22.3 s, post 73.7 ± 67.2 s, *p* = 0.033)-IPAQ (LS: baseline 1427.1 ± 1849.6 METs, post 2731.4 ± 1966.9 METs, *p* = 0.055; VR: baseline 1695.7 ± 1410.1 METs, post 3212.1 ± 1972.5 METs, *p* = 0.011; WP: baseline 1796.1 ± 1525.0 METs, post 3030.7 ± 1382.0 METs; *p* = 0.09)

## 4. Discussion

The main findings were that changes in anthropometric, cardiovascular, and muscle fitness variables occurred as a result of training programs performed: (1) under the supervision of an experienced trainer during a livestream, (2) without supervision following a video recording, or (3) without supervision following a written program. The results obtained from the whole sample show how remote online training effectively improves both muscle and cardiovascular fitness. In addition, the physical activity levels of all subjects also increased. While improvements in muscle fitness were reported in all three groups, improvements in cardiovascular variables were seen more so in the livestream group under the supervision of the experienced trainer. This aspect could be unexpected; from the supervision, we expect correct execution of the exercise and, therefore, a proper pattern of muscle contraction, resulting in increased muscle fitness. The results obtained on cardiovascular variables induce possible speculation regarding the role of supervised exercise, even remotely; correctly undertaking an exercise program results in further improvements. It is also likely better for management of training times in terms of the alternating effort–recovery of the entire workout. The lack of variation in anthropometric parameters, such as weight and BMI, could be caused by the lack of nutritional indications, the duration of the intervention, and the initial condition of the normal weight of the sample.

Previous studies on populations with chronic diseases [7,8] did not perform direct assessments as in the present study, but reported the rate of adherence (% sessions attended), retention (% participants who completed the intervention), and safety (number of events). The adherence rate of the LS group is in line with the study by Winters-Stone et al. [7]; however, a further comparison of the results of the fitness assessments obtained from this study is not possible.

The coronavirus pandemic has promoted technology to offer online remote health services. The planning and execution of medical examinations concerning sports and exercise medicine were also carried out remotely [23]. Other health activities, such as physiotherapy, also implemented services realized through online proposals [24,25].

Online remote physical training was used before the pandemic; in 2020, it ranked 26th in the worldwide survey of fitness trends. However, in 2021, it ranked 1st due to a shift in the fitness market from clubs to homes due to the pandemic. In 2022, it is 9th, as although coronavirus is still present, the need for greater social interaction likely resulted in this change [9]. However, some people may still want to maintain social distance during workouts. Therefore, determining which remote training modalities are the most effective could have an impact on the training program chosen and may also help to achieve the best adherence rate.

The indications provided by the fitness sector show an apparent effect of the pandemic on the European health and fitness industry [26]. People are aware that physical activity is healthy, but not everyone can or wants to go to the gym. Furthermore, online distance learning could be the first approach for new customers. Cronshaw’s study [27] shows that those who have experienced the benefits of online distance training question whether they need to go back to a physical gym. Therefore, gym owners must develop another proposal, offering online classes to those who cannot or do not want to join the gym. Therefore, the market adapted by offering hybrid fitness services that have become a reality. However, this could compromise the ability to deliver such services by professionals accustomed to offering workouts in the traditional in-person modality. Considering long-term fitness consumption, the integrated digital–physical model can help different groups of people who face barriers when attending the gym.

This study has some limitations. Firstly, the reliability of the parameters regarding the self-reported adhesion of the unsupervised groups (VR and WP) must be considered. However, this aspect is entirely within the scope of the study: the increase in physical activity levels shows a continuous training engagement of the sample. Furthermore, compliance with the agreements between the experienced trainer and the subject plays a crucial role in the effectiveness of any training proposal. Secondly, the online remote training program did not include aerobic activity. The proposed program envisaged feasibility regardless of the use of specific tools. Different aerobic training tools (treadmill vs. stationary bike vs. rowing machine) may have been confounding factors in the results due to the different adaptations induced by different movement patterns. However, the results show that the proposed training program also managed to modify cardiovascular variables.

### Practical Implications

Remote online training has effectively become a way to deliver exercise. Therefore, the practitioner should consider this delivery model as an option for their clients.The three training methods proved to be effective. Therefore, the choice of delivery method should be based on the individual characteristics of the subjects and their preference.The livestreaming mode with the supervision of an experienced trainer was the most effective. Therefore, it should be preferred in the case of physical exercise on a single subject.

## 5. Conclusions

In summary, the present study results indicate that remote online training was effective in healthy young male adults for improving muscle and cardiovascular fitness. However, the livestreaming model with expert supervision appears to be the delivery modality that achieves the higher effects, especially on cardiovascular parameters.

Further studies in other subjects are needed in order to establish the effectiveness of remote online training in other study populations.

## Figures and Tables

**Table 1 sports-10-00170-t001:** Changes after remote training in anthropometric and cardiovascular variables, in muscle fitness, and physical activity level in the whole study sample of 21 subjects. Data are expressed as mean and S.D. for the results obtained before (PRE) and after (POST) training. ∆ variation between baseline and post; C.I. (95%) = 95% confidence intervals; E.S. = effect size.

		PRE	POST	∆	C.I. (95%)	E.S.	*p* Value
Anthropometric variables	Weight (kg)	73.4±8.7	73.2±8.6	−0.29	−1.4|0.8	0.023	0.579
	BMI (Kg/m^2^)	22.6 ± 2.3	22.5 ± 2.2	−0.10	−0.4|0.2	0.044	0.539
	Waist Circumference (cm)	83.6 ± 6.0	82.3 ± 5.7	−1.29	−2.1|−0.5	0.170	0.004
Muscle fitness	Sit and Reach (cm)	26.1 ± 8.6	29.0 ± 7.2	3.4	0.7|6.2	0.365	0.018
	Shoulder Flexion (°)	174.2 ± 7.0	185.5 ± 5.0	11.0	5.6|16.6	1.858	0.000
	Shoulder Extension (°)	60.5 ± 9.7	66.8 ± 8.8	4.6	−0.9|10.2	0.680	0.097
	Shoulder Internal Rotation (°)	56.4 ± 16.9	69.3 ± 10.1	15.0	5.7|24.3	0.926	0.003
	One Leg Test (s)	23.9 ± 18.0	55.5 ± 47.4	32.8	3.6|61.9	0.881	0.030
	Handgrip (kg)	88.7 ± 21.3	95.6 ± 19.3	5.9	0.4|11.5	0.339	0.036
	Maximal Push-up (rep.)	26.3 ± 10.1	34.0 ± 7.9	8.7	5.0|12.4	0.849	0.000
	Maximal Plank (s)	90.8 ± 26.7	117.4 ± 30.4	29.1	17.2|41.1	0.930	0.000
Cardiovascular variables	Rest HR (bpm)	69.0 ± 7.2	66.3 ± 4.1	−7.3	−10.6|−3.9	0.461	0.000
	Ruffier Index (bpm)	9.0 ± 2.2	7.9 ± 1.6	−2.1	−3.1|−1.2	0.572	0.000
Physical activity level	IPAQ (METs)	1639.6 ± 1531.6	2991.4 ± 1715.3	1304.3	360.4|2248.2	0.831	0.009

**Table 2 sports-10-00170-t002:** Effectiveness of the livestream (LS) delivery method for remote training. Data are expressed as mean and S.D. for the results obtained before (PRE) and after (POST) training. ∆ variation between baseline and post. C.I. (95%) = 95% confidence intervals; E.S. = effect size.

		LS					
		PRE	POST	∆	C.I. (95%)	E.S.	*p* Value
Muscle fitness	Shoulder Flexion (°)	175.4 ± 6.8	186.4 ± 6.4	11.1	3.4|18.7	1.666	0.000
	Handgrip (kg)	85.9 ± 22.0	91.8 ± 23.3	5.9	−1.8|13.7	0.260	0.358
	Maximal Push-up (rep.)	23.1 ± 12.1	31.9 ± 8.7	8.7	3.6|13.9	0.835	0.000
	Maximal Plank (s)	85.4 ± 27.3	114.6 ± 26.3	29.1	12.4|45.9	1.089	0.000
Cardiovascular variables	Rest HR (bpm)	73.4 ± 7.0	66.1 ± 3.8	−7.3	−11.9|−2.7	1.296	0.000
	Ruffier Index (bpm)	9.8 ± 2.1	7.7 ± 1.7	−2.1	−3.5|−0.8	1.099	0.000

**Table 3 sports-10-00170-t003:** Effectiveness of the video recording (VR) delivery method for remote training. Data are expressed as mean and S.D. for the results obtained before (PRE) and after (POST) training. ∆ variation between baseline and post. C.I. (95%) = 95% confidence intervals; E.S. = effect size.

		VR					
		PRE	POST	∆	C.I. (95%)	E.S.	*p* Value
Muscle fitness	Shoulder Flexion (°)	173.9 ± 8.3	185.7 ± 4.5	11.7	4.1|19.4	1.767	0.000
	Handgrip (kg)	90.8 ± 23.6	101.0 ± 17.7	10.16	2.4|17.9	0.489	0.002
	Maximal Push-up (rep.)	25.4 ± 10.5	33.0 ± 8.1	7.57	2.4|12.7	0.810	0.000
	Maximal Plank (s)	89.6± 27.1	111.1 ± 24.9	21.57	4.8|38.3	0.826	0.002
Cardiovascular variables	Rest HR (bpm)	68.6 ± 8.6	65.7 ± 5.7	−2.9	−7.5|1.8	0.397	1.000
	Ruffier Index (bpm)	9.8 ± 1.4	8.2 ± 1.2	−1.6	−2.3|−0.3	1.227	0.006

**Table 4 sports-10-00170-t004:** Effectiveness of the written program (WP) delivery method for remote training. Data are expressed as mean and S.D. for the results obtained before (PRE) and after (POST) training. ∆ variation between baseline and post. C.I. (95%) = 95% confidence intervals; E.S. = effect size.

		WP					
		PRE	POST	∆	C.I. (95%)	E.S.	*p* Value
Muscle fitness	Shoulder Flexion (°)	173.2 ± 6.9	184.3 ± 4.5	11.1	3.4|18.7	1.905	0.000
	Handgrip (kg)	89.3 ± 21.4	94.0 ± 18.2	4.7	−3.1|12.4	0.236	1.000
	Maximal Push-up (rep.)	30.3 ± 7.3	37.0 ± 7.1	6.7	1.6|11.9	0.930	0.002
	Maximal Plank (s)	97.4 ± 28.3	126.6 ± 40.2	29.1	12.4|45.9	0.840	0.000
Cardiovascular variables	Rest HR (bpm)	65.0 ± 2.4	67.1 ± 3.0	2.14	−2.5|6.8	0.773	1.000
	Ruffier Index (bpm)	7.5 ± 2.4	7.9 ± 2.1	0.46	−0.9|1.8	0.177	1.000

## Data Availability

The data of this study are available upon request to the corresponding author at the address gabriele.mascherini@unifi.it.

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
