# Peer review of "Effectiveness of Different Modalities of Remote Online Training in Young Healthy Males"

_sports, 2022, doi:10.3390/sports10110170_

Round 1

Reviewer 1 Report

General comments

The authors addressed a very important topic, i,e., the effectiveness of different modalities of remote online training  in young adults.

They compared the effectiveness of three different training programs: supervised live streaming, unsupervised following a video recording, and unsupervised following a written program.  The results are interesting.

In my opinion, the manuscript is well-written.

I only make the following comments to the authors.

Specific comments

Abstract

The abstract is well-structured and clearly understandable.

Key Words are relevant.

Introduction

The authors summarized the literature correctly.

Materials and Methods

The methodology is clearly explained.

The research design is rigorous.

The instruments are validated and reliable.

The statistics are correct.

Results

The results are written correctly.

Discussion

The authors' conclusions are justified.

The take-home message is clear.

The limitations are addressed.

Author Response

Reviewer 1

The authors addressed a very important topic, i,e., the effectiveness of different modalities of remote online training  in young adults.

They compared the effectiveness of three different training programs: supervised live streaming, unsupervised following a video recording, and unsupervised following a written program.  The results are interesting.

In my opinion, the manuscript is well-written.

I only make the following comments to the authors.

Specific comments

Abstract

The abstract is well-structured and clearly understandable.

Key Words are relevant.

Introduction

The authors summarized the literature correctly.

Materials and Methods

The methodology is clearly explained.

The research design is rigorous.

The instruments are validated and reliable.

The statistics are correct.

Results

The results are written correctly.

Discussion

The authors' conclusions are justified.

The take-home message is clear.

 The limitations are addressed.

The authors would like to thank the reviewer for appreciating our work and suggestions provided to improve our manuscript. Changes made to the manuscript are highlighted in red. Below are the answers to the reviewer’s comments.

Reviewer 2 Report

I find the study interesting, and quite up to date, as it deals with research on quite up-to-date methodologies. However, it can be considerably improved. 

It is not adapted to the journal format: separate paragraphs, tables are not formatted, the table title is below, instead of above, , 

Line 43-44, which studies? Cite. 

The introduction is very short, before covid there was nothing on this topic... Why did only boys participate?

Why did only boys participate, why was the inclusion criterion used 18-25, and why was the sample size 18-25? From the characteristics of the sample, the participants seem to be in apparently good shape (22.6 kg/m2). In this case, I think it would be appropriate to adapt the title to include at least the sex of the participants, or even "healthy male".

Is the sample representative? A sample calculation is recommended to assess the representativeness of the results. 

How and from where were the participants recruited, how was the process?

Lines 89 to 93 should be placed after the participants.

The section mentions "study design" but does not specify what type of study it is.

What was the weight measured with?

Lines 119 and 124 "According to the 119 procedures" repetitive. Modify wording. 

One of the inclusion criteria was to have 2 hours a day, however the programme only runs three days a week. Perhaps their criteria was too restrictive, as the programme only needs these 2 hours in 3 days. 

Table 2, include units of measurement. 

Discussion:

There is a lack of studies that and references training programmes under similar conditions that have obtained similar or different results and justification of how theirs differs from the rest. 

The discussion is very brief and poor, their results are hardly discussed with others of similar characteristics. 

Do you think these results would be the same if the study were carried out in another, less uniform population? 

Practical implications should be included at the end of the discussion section. 

The conclusions should respond to your objective, i.e. what has been obtained from the described objective, what happened with the variables "anthropometric and cardiovascular variables, muscle fitness, and physical activity levels". The conclusions are very poor and do not allow the reader to know the real conclusions of the study. 

Author Response

Reviewer 2

I find the study interesting, and quite up to date, as it deals with research on quite up-to-date methodologies. However, it can be considerably improved. 

The authors would like to thank the reviewer for appreciating our work and suggestions provided to improve our manuscript. Changes made to the manuscript are highlighted in red. Below are the answers to the reviewer’s comments.

It is not adapted to the journal format: separate paragraphs, tables are not formatted, the table title is below, instead of above, , 

Answer: Thanks for the comment. The paragraphs have been separated and numbered, especially in the methods section. Table has been formatted and titles were moved above.

Line 43-44, which studies? Cite. 

Answer: Thanks for the comment. The citation is number 6 which is already inserted in both the previous and the next sentence. For clarity, ref. 6 is also inserted at the end of the sentence at line 44.

The introduction is very short, before covid there was nothing on this topic... Why did only boys participate?

Why did only boys participate, why was the inclusion criterion used 18-25, and why was the sample size 18-25? From the characteristics of the sample, the participants seem to be in apparently good shape (22.6 kg/m2). In this case, I think it would be appropriate to adapt the title to include at least the sex of the participants, or even "healthy male".

Answer: Thanks for the comment and suggestion. Only male subjects were enrolled in the present research in order to make a more appropriate comparison with the existing literature, as previous studies focused on male population only (McNamara, J. M., Swalm, R. L., Stearne, D. J., and Covassin, T. M. (2008). Weight training online. Journal of Strength and Conditioning Research, 22 (4), 1164– 1168. https: //doi.org/10.1519/JSC.0b013e31816eb4e0). 

Two sentence and two references has been added in introduction and the title has been modified as “Effectiveness of different modalities of remote online training in young healthy male”

Is the sample representative? A sample calculation is recommended to assess the representativeness of the results. 

Answer: We thank the reviewer for the interesting and important comment. We agree that researchers are often interested in making inferences about fixed effects in a linear multilevel mixed-effects model and the test statistics for testing hypotheses about fixed effects in balanced split and repeated-measures designs have exact t or F distributions. For large samples, the null sampling distributions of the test statistics can be approximated by a normal distribution for a one-hypothesis test and a χ2 distribution for a multiple-hypotheses test. However, these large-sample approximations may not be appropriate in small samples, and t and F distributions may provide better approximations. That is why we fitted the models using the maximum residual likelihood for accounting for the small sample. Furthermore, the repeated measures ANOVA method was used for computing the degrees of freedom of a t distribution, as subjects were tested before and after the training protocols, alongside the contrast method tests (ANOVA-style tests). Furthermore, to avoid type 1 error, we subsequently used Bonferroni post-hoc tests for multiple comparison adjustments across all terms when significant main effects and interactions were found, by therefore adjusting the significance threshold at p-value < 0.003. Although we recognize that this procedure might have increased the probability of type 2 error, we are likely confident that the sample was representative of the results. In fact, this assumption is confirmed by a previous study (Bell B.A., et al. 2010. SAS Global Forum 2010) in which the authors demonstrated that level-1 predictors reach a mean at or above 0.80 power estimates when a linear multilevel mixed-effects model with one level had sample size between 20-40 (our sample was of 21 subjects). We hope that the explanation provided fulfills the request made by the reviewer. Meanwhile we thank the reviewer again for the comment.

How and from where were the participants recruited, how was the process?

Answer: Thanks for the comment. A sentence has been added at the beginning of 2.1 Participants and study design:

“The recruitment process involved the dissemination of the initiative among university students. Those who: 1) gave their availability, 2) guaranteed the achievement of the sample size, 3) met the inclusion criteria and 4) signed informed consent, were enrolled in the study.”

Lines 89 to 93 should be placed after the participants.

Answer: Thanks for the comment. The sentences 89-93 were moved accordingly. 

The section mentions "study design" but does not specify what type of study it is.

Answer: Thanks for the comment. The sentence “following the model for randomized studies with parallel groups” has been added after the participants' description.

What was the weight measured with?

Answer: Thanks for the comment. The tools of  Seca GmbH & Co. were used for both weight and height. The model (786) and the country (Germany) has been added in the text.

Lines 119 and 124 "According to the 119 procedures" repetitive. Modify wording. 

Answer: Thanks for the comment. This wording has been removed.

One of the inclusion criteria was to have 2 hours a day, however the programme only runs three days a week. Perhaps their criteria was too restrictive, as the programme only needs these 2 hours in 3 days. 

Answer: Thanks for the comment. The authors agree with the reviewer. However, this inclusion criterion was chosen to reduce the risk of leaving the group during the study period. In addition, this provided the possibility for those who followed the live stream not to be bound to certain days of the week.

Table 2, include units of measurement. 

Answer: Thanks for the suggestion. The addition has been made accordingly.

Discussion:

There is a lack of studies that and references training programmes under similar conditions that have obtained similar or different results and justification of how theirs differs from the rest. 

The discussion is very brief and poor, their results are hardly discussed with others of similar characteristics. 

Answer: Thanks for the comment. Currently, there are few articles available that have dealt with the topic of remote physical training in healthy young male adults. To the author's knowledge, the references list contains the few who have dealt with this topic. Therefore, it does not seem possible to make a comparative discussion with other literature data. In particular, the studies carried out on pathological subjects (ref. 7, 8) did not carry out direct evaluations like the present study, but reported Adherence (% sessions attended), retention (% participants who completed the intervention) and safety (# adverse events) making it difficult to compare with the results obtained from this study. For this reason, the authors in paragraphs 3-4-5 of the discussion section commented on their results trying to contextualize them to the historical period, the fitness market and their relevance for the health of the population.

In the author’s point of view, this aspect appears to be a strength and an innovation of this study.

Do you think these results would be the same if the study were carried out in another, less uniform population? 

Answer: Thanks for the comment and the suggestion. The authors are interested in this aspect; therefore, a sentence has been added in the conclusions section.

“Further studies in other kinds of subjects are needed in order to establish the effectiveness of remote on line training in other study populations”.

Practical implications should be included at the end of the discussion section. 

Answer: Thanks for the comment. This section has been moved accordingly.

The conclusions should respond to your objective, i.e. what has been obtained from the described objective, what happened with the variables "anthropometric and cardiovascular variables, muscle fitness, and physical activity levels". The conclusions are very poor and do not allow the reader to know the real conclusions of the study. 

Answer: Thanks for the comment. The conclusions section has been improved. Now the section is:

“In summary, the present study results indicate that remote online training was effective in healthy young male adults for improving muscle and cardiovascular fitness. However, live streaming mode, with expert supervision, appears to be the delivery modality that achieves the higher effects, especially on cardiovascular parameters.

Further studies in other kinds of subjects are needed in order to establish the effectiveness of remote on line training in other study populations.”

Reviewer 3 Report

Take a look at the comments.

Author Response

Reviewer 3

I have carefully reviewed manuscript of Daveri et al. titled: " Effectiveness of different modalities of remote online training in young adults ".

The authors would like to thank the reviewer for the suggestions provided to improve our manuscript. Changes made to the manuscript are highlighted in red. Below are the answers to the reviewer’s comments.

Please check the comments for your manuscript:

  • Introduction

o Please try to find flow in introduction. Everything is written up and down and it is confusing – try to get direct into problem and rationale after good background base

Answer: Thanks for the comment. Two sentences have been added to the introduction section to make it more understandable: 1. the hypothesis of the study has been added and 2. two studies relating to remote online training in subjects with chronic diseases have been introduced, thus also increasing the number of references.

o Do you have hypothesis?

Answer: Thanks for the comment. A sentence has been added at the end of the introduction section: “The hypothesis of this study is that the supervision of the instructor can provide greater improvements even for remote online training.”

o You can’t have only 7 references for introduction!

o You must go and dig heavily into the problem you want to write about…there are many COVID-19 references which write about exercise and pandemic.

Answer: Thanks for the comment. Currently, there is not a lot of research that deals specifically with remote online training. Much research is focussed on other aspects of physical activity during COVID-19 without a close link to online training (the authors also published about the relationship between physical activity and COVID-19). 

Other publications available on the feasibility of online training and COVID are related to diseased populations such as cancer, multiple sclerosis: a new sentence has been added in the introduction section with 2 new references.

This aspect seems to be a strength and innovation of this study that could expand the literature on remote online training.

  • Methods

o What are the countries of manufacturer of equipment you have used?

Answer: Thanks for the comment. The tools of  Seca GmbH & Co. were used for both weight and height. The model and the country (786, Germany) has been added in the text.

o Why do you use term muscle fitness?

â–ª What is definition of it

â–ª References

Answer: Thanks for the comment. The muscle fitness is defined as “integration of muscle strength, muscle endurance and flexibility, and is an integral portion of total health related fitness”. This definition was provided by the American College of Sports Medicine. Position Stand: The recommended quantity and quality of exercise for developing and maintaining cardiorespiratory and muscular fitness, and flexibility in healthy adults. Med Sci Sports Exerc. 1998;30:975-991. This reference has been added as number 10.

â–ª Also make this subtitle more organised and easier to read

Answer: Thanks for the comment. This paragraph has been made easier to follow through the use of numbered bulleted lists.

  • Results

o Tables’ titles should go above the table. Please check author guidelines

Answer: Thanks for the suggestion. The addition has been made accordingly.

o Tables are confusing and hard to understand without legends

Answer: Thanks for the suggestion. The legends about the missing abbreviations have been added accordingly.

  • Discussion

o What do you write here about? You don’t write about your results of the study

o Discussion is so weak – firstly write about your main findings

o Where are references to similar studies?

o Where are references to support your findings?

Answer: Thanks for the comment. Currently, there are few articles available that have dealt with the topic of remote physical training. To the author's knowledge, the references list contains the few who have dealt with this topic:

- n. 6 used in the introduction;

- n.7-8 has been added in introduction section, about feasibility of online training in diseased subjects; 

- n. 21 about evaluation in sports medicine, used in the discussion;

- n. 22 about physiotherapy, used in the discussion;

- n.23 on exercise in the older adults used in the discussion;

- n. 25 about remote exercises for gyms.

Therefore, it does not seem possible to make a comparative discussion with other literature data. In particular, the studies carried out on pathological subjects (ref. 7, 8) did not carry out direct evaluations like the present study, but reported Adherence (% sessions attended), retention (% participants who completed the intervention) and safety (# adverse events) making it difficult to compare with the results obtained from this study. For this reason, the authors in paragraphs 3-4-5 of the discussion section commented on their results trying to contextualize them to the historical period, the fitness market and their relevance for the health of the population.

In the author’s point of view, this aspect appears to be a strength and an innovation of this study.

In order to get a more solid discussion, the "Practical Implications" have been moved as 4.1 in the final part of the discussion section.

  • Conclusions

o Conclusion is weak…write about the results of the study and reorganise this

Answer: Thanks for the comment. The conclusions section has been improved. Now the section is:

“In summary, the present study results indicate that remote online training was effective in healthy young male adults for improving muscle and cardiovascular fitness. However, live streaming mode, with expert supervision, appears to be the delivery modality that achieves the higher effects, especially on cardiovascular parameters.

Further studies in other kinds of subjects are needed in order to establish the effectiveness of remote on line training in other study populations.”

Round 2

Reviewer 2 Report

The authors have addressed the suggested comments

Author Response

The authors would like to thank the reviewer for appreciating our work and suggestions provided to improve our manuscript.

Reviewer 3 Report

Dear authors,

some parts of paper are improved, but why is there no changes in discussion - even tough you wrote that you changed/improved. Why is only no tracked change in discussion?

Kind regards

Author Response

Dear authors,

some parts of paper are improved, but why is there no changes in discussion - even tough you wrote that you changed/improved. Why is only no tracked change in discussion?

Kind regards

Answer: Thanks for the comment. Taking up the comments from the previous round of review, we further elaborated the Discussion section:

o What do you write here about? You don’t write about your results of the study

o Discussion is so weak – firstly write about your main findings

According to the comments, the main findings of this study have been moved and commented at the beginning of the discussion section. The modified part is colored red in the second revision version.

o Where are references to similar studies?

o Where are references to support your findings?

As indicated in the first round of review, literature focusing on  remote online training with direct assessments of physical fitness is limited, as studies mostly use questionnaires or feasibility assessments. Therefore, the discussion section has been modified accordingly:

“Previous studies on populations with chronic diseases [7, 8] did not perform direct assessments as in the present study, but reported the rate of adherence (% sessions attended), retention (% participants who completed the intervention) and safety (number of events): adherence rate of the LS group is in line with the study by Winters-Stone et al. [7]; however, a further comparison of the results of the fitness assessments obtained from this study is not possible.”

Round 3

Reviewer 3 Report

Let the editors decide.

Thank you for answers.